# Intracellular Growth and Cell Cycle Progression are Dependent on (p)ppGpp Synthetase/Hydrolase in *Brucella abortus*

**DOI:** 10.3390/pathogens9070571

**Published:** 2020-07-14

**Authors:** Mathilde Van der Henst, Elodie Carlier, Xavier De Bolle

**Affiliations:** Unité de Recherche en Biologie des Microorganismes (URBM), University of Namur, 61 rue de Bruxelles, 5000 Namur, Belgium; mathilde.vanderhenst@unamur.be (M.V.d.H.); elodie.carlier@unamur.be (E.C.)

**Keywords:** Brucella, cell cycle, (p)ppGpp, *rsh*

## Abstract

*Brucella abortus* is a pathogenic bacterium able to proliferate inside host cells. During the first steps of its trafficking, it is able to block the progression of its cell cycle, remaining at the G1 stage for several hours, before it reaches its replication niche. We hypothesized that starvation mediated by guanosine tetra- or penta-phosphate, (p)ppGpp, could be involved in the cell cycle arrest. Rsh is the (p)ppGpp synthetase/hydrolase. A *B. abortus* ∆*rsh* mutant is unable to grow in minimal medium, it is unable to survive in stationary phase in rich medium and it is unable to proliferate inside RAW 264.7 macrophages. A strain producing the heterologous constitutive (p)ppGpp hydrolase Mesh1b is also unable to proliferate inside these macrophages. Altogether, these data suggest that (p)ppGpp is necessary to allow *B. abortus* to adapt to its intracellular growth conditions. The deletion of *dksA*, proposed to mediate a part of the effect of (p)ppGpp on transcription, does not affect *B. abortus* growth in culture or inside macrophages. Expression of a gene coding for a constitutively active (p)ppGpp synthetase slows down growth in rich medium and inside macrophages. Using an mCherry–ParB fusion able to bind to the replication origin of the main chromosome of *B. abortus*, we observed that expression of the constitutive (p)ppGpp synthetase gene generates an accumulation of bacteria at the G1 phase. We thus propose that (p)ppGpp accumulation could be one of the factors contributing to the G1 arrest observed for *B. abortus* in RAW 264.7 macrophages.

## 1. Introduction

Bacteria from the *Brucella* genus are the causative agents of brucellosis, a neglected disease which constitutes a worldwide anthropozoonosis. *Brucella* spp. are Gram negative alphaproteobacteria belonging to the Rhizobiales order [1]. *Brucella abortus* causes severe symptoms in mammals such as abortion in pregnant females and sterility in males. In humans, the disease is characterized by an undulant fever, also named Malta fever, and in the long term the infection leads to chronicity and symptoms such as arthritis, endocarditis and can have a fatal outcome without treatment [1]. In their hosts, *Brucellae* invade, survive and replicate inside professional and non-professional phagocytic cells such as macrophages and trophoblasts. Inside host cells, *Brucellae* are found in vacuoles named *Brucella* containing vacuoles (BCVs). In the first part of their trafficking, they successively harbor markers of early and late endosomes, a phase of the trafficking in which the bacterium does not proliferate [2,3]. This compartment presents a pH of about 4.0 to 4.5 and this acidification is essential for the successful establishment of *B. suis* infection [4]. Afterwards, the bacteria are found in BCVs having markers of the endoplasmic reticulum, where they replicate [5]. Later in the cellular infection, bacteria are found in vacuoles characterized by autophagy-related proteins [6].

Different cellular models for in vitro study of *B. abortus* infection of have been developed, such as the use of RAW 264.7 macrophages and HeLa epithelial cells. Some years ago, the investigations of the *B. abortus* infection process in these models revealed that the cell cycle regulation of *B. abortus* is linked to its virulence [2]. Indeed, for bacteria that did not segregate duplicated replication origins, the so-called G1 cells are more infectious than the S or G2 phase bacteria, i.e., bacteria currently replicating their genome or at the stage between the completion of genome replication genome and cell vision, respectively. More importantly, after internalization, bacteria remain in the G1 stage for up to 8 h, depending the host cell type, in Lamp1-positive compartments before reaching the endoplasmic reticulum where *B. abortus* can restart its cell cycle, its DNA replication and actively proliferate [2]. During the first hours of the infection, in BCVs with endocytic markers, *B. abortus* encounters harsh conditions such as acidic stress [7] and alkylating stress [8]. In addition, it was already proposed that *B. abortus* has to face a starvation conditions inside host cells [9]. Starvation is the most obvious condition that could explain why *B. abortus* is blocked at the G1 stage of the cell cycle during the first phase of its intracellular trafficking in HeLa cells and RAW 264.7 macrophages. Starvation sensing is classically involving the synthesis of (p)ppGpp (guanosine penta- or tetra-phosphate), also called alarmone. The synthesis and degradation of (p)ppGpp are catalyzed by enzymes of the RelA/SpoT family, also called Rsh enzymes. It was found that *rsh* mutants, which should be not able to produce (p)ppGpp anymore, are strongly impaired during in vitro infection as well as during murine infection [10,11].

The alarmone (p)ppGpp is widely used by bacteria to quickly adapt to stress conditions such as nutrient starvation. The production and accumulation of this alarmone induces pleiotropic effects, modulating transcription and translation, that commonly result in cell cycle and DNA replication delay [12,13,14,15]. The ability to produce (p)ppGpp has been associated with virulence in bacterial pathogens belonging to relatively distant phylogenetic groups, such as *Legionella pneumophila* [16], *Vibrio cholerae* [17], and *Mycobacterium tuberculosis* [18]. In *Escherichia coli*, during the stringent response induced by starvation, (p)ppGpp binds directly to a site located at the interface between the β’ and ω subunits of the RNA polymerase [19]. A second distinct site between the β’ subunit and the DksA transcription factor has been shown to be bound by (p)ppGpp as well [20]. This interaction has been shown to enhance the transcriptional effects of DksA on the RNA polymerase, suggesting synergistic effects of DksA and (p)ppGpp together [20].

The RelA/SpoT homolog proteins are responsible for (p)ppGpp homeostasis. In *E. coli*, there are two enzymes of the Rsh family, RelA and SpoT [21]. SpoT contains a synthetase domain, a hydrolase domain, and two C-terminal regulatory domains; thus, this enzyme can both catalyze the production and the degradation of (p)ppGpp, respectively. RelA contains similar domains, however the functionality of the hydrolase domain of RelA has been lost during evolution, leading to a monofunctional enzyme that can only synthesize the alarmone [22]. In most alphaproteobacteria, including *B. abortus*, the production and the degradation of (p)ppGpp depends on one enzyme named Rsh (for RelA SpoT homolog) [10,21].

In the present study, we analyzed the impact of alterations in (p)ppGpp synthesis or degradation on the growth, the cell cycle and the infection process of *B. abortus*. We show that mutants either unable to produce (p)ppGpp or producing a (p)ppGpp hydrolase are impaired for the infection process. In addition, our results show that expression of a constitutive (p)ppGpp synthetase negatively impacts growth and DNA replication of *B. abortus*, and also leads to a strong proliferation defect during infection of RAW 264.7 macrophages. We also observed that a *B. abortus dksA* null mutant was able to proliferate inside host cells as the wild type (WT) strain, suggesting that DksA is not crucially involved in the (p)ppGpp-dependent phenotypes observed during infection. These results suggest that adjustment of (p)ppGpp levels are crucial for the infection process in *B. abortus*.

## 2. Results

### 2.1. rsh Deletion Drastically Impacts Growth in Minimal Medium and the Infection Process

We generated a Δ*rsh* strain by allelic replacement in *B. abortus* 544 and we assayed the growth of this strain in rich culture medium (2YT) as well as in Plommet minimal medium [23] supplemented with erythritol as a carbon source. The growth of Δ*rsh* in 2YT was similar to the WT strain during the exponential phase, but the shift into the stationary phase occurred later and at a higher optical density (OD) compared to the wild type strain (Figure 1A). To evaluate bacterial viability, we counted the colony forming units (CFUs) throughout the culture in liquid rich medium (Figure 2). The Δ*rsh* strain showed a clear survival defect during the stationary phase, marked by a decrease in CFUs between 24 h and 48 h compared to the WT and the complemented strain.

Because it has been shown in other bacteria that the stringent response is linked to nutrient availability, we tested the growth of the Δ*rsh* strain in Plommet minimal medium, supplemented with erythritol as a carbon source, to mimic starvation conditions. The Δ*rsh* showed a clear growth defect compared to the WT as the OD rapidly decreased during the mid-exponential phase, indicating that Δ*rsh* cannot grow and survive in this medium, as expected for a mutant unable to produce (p)ppGpp (Figure 1B).

We tested the ability of Δ*rsh* to infect and multiply inside RAW 264.7 macrophages compared to the WT strain and the complemented strain by performing CFU counting throughout the cellular infection (Figure 3). The Δ*rsh* mutant showed a significant decrease in CFUs at 24 h post-infection compared to the WT strain, suggesting that the *rsh* gene is required for intracellular proliferation. A slight but significant difference was also observed at 2 h post-infection between the WT and complemented strain (Figure 3), probably highlighting a low toxicity of the vector or a *rsh* overexpression effect.

### 2.2. The Artificial Hydrolysis of (p)ppGpp Leads to a Δrsh Phenotype during Infection

Since Rsh is responsible for (p)ppGpp homeostasis and an Δ*rsh* mutant failed to proliferate inside RAW 264.7 cells, we tested the involvement of (p)ppGpp in the infection process. However, it is known that Rsh is involved in regulation networks through protein–protein contacts [14,24] in other bacteria. Therefore, we cannot rule out that the absence of the Rsh protein, rather than the absence of (p)ppGpp, would be responsible for the defect observed in infection. This is reinforced by the observation that mutants for homologs of the glutamine-dependent control pathway of Rsh are also attenuated in RAW 264.7 macrophages [25]. We thus generated a strain in which (p)ppGpp is hydrolyzed by a strong (p)ppGpp hydrolase, a product of the *mesh1* gene from *Drosophila melanogaster* [26]. Indeed, it was shown that Mesh1 was active in vitro and in vivo [26]. We thus expect this heterologous enzyme to be constitutive in *B. abortus*. We adapted the *mesh1* coding sequence for the codon bias of *B. abortus* and expressed the resulting coding sequence on a medium copy replicative plasmid, leading to the *B. abortus* pBBRi-*mesh1b* strain. Interestingly, this strain showed a clear decrease in CFUs at 24 h post-infection of RAW 264.7 macrophages (Figure 4), which is consistent with a crucial role played by (p)ppGpp to allow growth inside host cells, as suggested above.

### 2.3. Expression of a Constitutive Allele for a (p)ppGpp Synthetase Impacts Bacterial Growth and Chromosome Replication

In order to get more insight about the role of (p)ppGpp in *B. abortus*, we constructed a strain that artificially produces this alarmone. We used a truncated version of the *relA* gene from *E. coli*, *relA’* [12] that removes the C-terminal regulatory domains of the encoded protein. The *relA’* coding sequence was inserted downstream of an isopropyl *β*-D-1-thiogalactoside (IPTG)-inducible promoter on the pSRK replicative plasmid [27]. The resulting strain, named *pSRK-relA’*, is supposed to produce (p)ppGpp synthetase when IPTG is added to the medium. As a negative control, we used the *pSRK-relA*’* strain containing the point mutation E335Q, which leads to a catalytically dead protein. Since the detection of (p)ppGpp levels using ^32^P is not compatible with our biosafety level 3 set up, we tried to gain indirect evidence that (p)ppGpp is indeed produced when the expression of *relA’* is induced. We assayed the growth of the *pSRK-relA’* and *pSRK-relA’** strains in rich culture medium with or without IPTG induction. The *pSRK-relA’*, *pSRK-relA*’* and WT strains grew equally in 2YT; however, when IPTG was added to the medium, a growth delay was only observed for the *pSRK-relA’* strain (Figure 5). This observation is consistent with the production of (p)ppGpp levels that are sufficient to limit growth when *relA’* expression is induced.

Since it was already reported that (p)ppGpp has an impact on DNA replication in *C. crescentus* and *E. coli* [12,13,14,15], we took advantage of a *B. abortus* strain allowing us to monitor the chromosomal replication status at the single cell level in order to study the impact of (p)ppGpp overproduction on DNA replication. This strain expresses an *mCherry-parB* allele that allows us to highlight the segregation of replication origin(s) of chromosome I. In this strain, one mCherry focus means that segregation has not yet started and the bacterium is probably in the G1 phase of the cell cycle, and two mCherry foci correspond to two segregated replication origins, meaning that the bacterium has already started replication and is thus in the S or G2 phase of the cell cycle [2]. The *pSRK-relA’* and *pSRK-relA’** plasmids were inserted in a *B. abortus mCherry-parB* strain and we counted the number of G1 bacteria every two hours for 6 h after the inoculation of bacteria in rich medium, with or without IPTG. Interestingly, we observed an increase in the proportion of G1 bacteria over the time of induction with IPTG for the *pSRK-relA’* strain (Appendix A). The proportion of G1 bacteria of the non-induced *pSRK-relA’* and both the induced or non-induced *pSRK-relA’** remained stable after the addition of IPTG (Figure 6). These results strongly suggested that artificial induction of (p)ppGpp synthesis could delay the transition between the G1 phase to the S phase and subsequently have an impact on the initiation of chromosomal replication in *B. abortus*.

### 2.4. Induced Production of a Constitutive (p)ppGpp Synthetase Leads to a Proliferation Defect during Infection

Since (p)ppGpp overproduction seemed to have an impact on replication, i.e., an increase of the proportion of G1 cells in the bacterial population, and that the G1 bacteria are more infectious, we decided to investigate the effect of overproduction of (p)ppGpp on the infection process. We infected RAW 264.7 macrophages with the *pSRK-relA’* strain induced or not with IPTG (Figure 7). The IPTG was kept in the cell culture medium during the infection for the induced condition. We first observed that bacterial internalization is not enhanced by the increase in the proportion of bacteria in the G1 phase of the cell cycle. We also observed that induction of *pSRK-relA’* induced a strong defect in intracellular proliferation compared to the WT and uninduced *pSRK-relA’* conditions. This result suggests that overproduction of (p)ppGpp during infection prevents growth in the intracellular niche. 

### 2.5. DksA Is Not Required during the Infection Process

Because (p)ppGpp seemed important during host infection and DksA is involved in a part of the (p)ppGpp transcriptional response in other species, we tested the ability of Δ*dksA* to infect and proliferate inside RAW 264.7 macrophages. No difference in CFUs was observed between WT and Δ*dksA* strains (Appendix A), meaning that DksA is not crucially involved in the infection process and that the phenotype observed for (p)ppGpp-deprived mutants (Δ*rsh* and *pBBRi-mesh1b*) is probably not mediated by DksA.

## 3. Discussion

*Brucella abortus* is able to control its cell cycle progression when it is inside host cells, particularly the replication and segregation of its replication origins [2]. However, the molecular mechanisms involved in this control are unknown. Since convincing data show that the ability to adapt to starvation is a key factor for the success of cellular infections by *Brucella melitensis* and *Brucella suis* [10,11], we further investigate the role of the (p)ppGpp, the alarmone produced in the presence of starvation conditions, and the Rsh enzyme that is proposed to synthesize this alarmone. We first confirmed that in *B. abortus* 544, like in other *Brucella* strains, Rsh is crucial for the success of a cellular infection (Figure 3), for the survival in stationary phase (Figure 2) and the growth in minimal medium (Figure 1). In agreement with the absence of proliferation of the *rsh* mutant in macrophages, a strain constitutively producing a (p)ppGpp hydrolase (Mesh1b) from *Drosophila melanogaster* is also unable to grow in RAW 264.7 macrophages. We observed that the survival of *pBBR-mesh1b* strain is less severely impacted during infection than the Δ*rsh* strain. One could imagine that this intermediate phenotype is due to the presence of residual alarmone in the *pBBR-mesh1b* strain, while it is not the case in the Δ*rsh* strain since the (p)ppGpp synthetase domain is not present. Another explanation for these different phenotypes could be that Rsh plays additional role(s) for survival in infection than the regulation of (p)ppGpp homeostasis. Interestingly, a mutant overproducing (p)ppGpp is also unable to proliferate in these cells, suggesting that the (p)ppGpp level should be in a specific range of concentration to allow cellular infection; having too much or not enough (p)ppGpp would be detrimental for the success of the cellular infection.

What does (p)ppGpp control and how is Rsh regulated? It was shown that the absence of *rsh* in *B. suis* affects the transcription of genes known to be involved in virulence [28], such as *pyrB*, which was shown to be essential for *B. abortus* proliferation in RAW 264.7 macrophages [25]. It is thus likely that a minimum level of (p)ppGpp would be required for the success of a cellular infection by *B. abortus*. It was shown that the glutamine pool modulates Rsh (called SpoT) in the model alpha-proteobacterium *Caulobacter crescentus* through the phosphotransferase system (PTS) [14]. Interestingly, mutants for components of this system were found to be attenuated in RAW 264.7 macrophages [25]. Moreover, PTS and the two-component regulator BvrR control the expression of the *virB* operon [29,30], coding for a type IV secretion system that is crucial for intracellular proliferation in most cell types [31]. These data indicate that a quite complex regulation network is probably linking Rsh control and virulence. However, the molecular mechanisms controlling Rsh activity in *B. abortus* are unknown and deserve further investigation.

One striking conclusion of our data is the moderate effect that (p)ppGpp overproduction has on the proportion of bacteria at the G1 stage of the cell cycle. Indeed, while overproduction seems to be sufficient to impair growth inside host cells (Figure 7), the proportion of G1 after 6 h of induction is about a third of the culture while it is approximately 15% in the absence of induction (Figure 6). Since the cell cycle takes about 3 h in the conditions tested, it is likely that only a fraction of the bacteria arrested their cell cycle at the G1 stage. Inside host cells, the proportion of G1 cells is about 75% and remains stable for 2 to 4 h at least [2], suggesting that other mechanisms are probably involved in the control of the cell cycle in host cells, early in the trafficking. These mechanisms could involve the acidic nature of the BCV, or the diffusion sensing proposed to occur through a regulation system homologous to quorum sensing [32]. More investigations are thus needed to discover the multiple factors involved in the cell cycle control of *B. abortus* inside host cells.

## 4. Materials and Methods

### 4.1. Strains and Growth Conditions

The reference strain *B. abortus* 544 was used for all experiments and was grown on solid or in liquid 2YT medium (LB 32 g/L Invitrogen, Yeast Extract 5g/L, BD and Peptone 6 g/L, BD) at 37 °C. *E. coli* strain DH10B was used for plasmid constructs and the conjugative strain *E. coli* S17-1 was used for mating with *B. abortus*. Both strains were cultivated in LB medium (Luria Bertani, Casein Hydrolysate 10g/L, NaCl 5g/L, Yeast Extract 5g/L) at 37 °C. Depending on the plasmid used, different selection antibiotics were added to the culture medium: ampicillin (100 μg/mL); carbenicillin (100 μg/mL); kanamycin (50 μg/mL for the replicative plasmid, and 10 μg/mL for the integrated plasmid); nalidixic acid (25 μL/mL); chloramphenicol (20 μg/mL for the replicative plasmid and 4 μg/mL for the integrated plasmid). Isopropyl β-d-1-thiogalactopyranoside (IPTG) was used at a concentration of 1 mM in bacterial liquid culture and at 10 mM in the mammalian cell culture medium during cellular infections. When the ∆*rsh* mutant was constructed, we added casamino acids 0.5% (BactoTM Casamino Acids from Thermo Fisher, Waltham, MA, USA) in the conjugation medium.

### 4.2. Strains Construction

Deletion strains were constructed by allelic exchange using the pNPTS138 vectors (M. R. K. Alley, Imperial College of Science, London, UK) carrying the upstream and the downstream regions of the targeted gene. The primer sequences used for amplification of the upstream region of the *dksA* gene were 5’-ttGGATCCcaagcgccagatcttca-3’ and 5’-ttGAATTCttcactcattctgaatcacccc-3’. The primer sequences used for amplification of the downstream region of the *dksA* gene were 5’-ttGAATTCtgatatcgaataatggtttggaaa-3’ and 5’-ttAAGCTTcgcccagcttcaaattac-3’.

We used the *rsh* deletion plasmid pMQ203 (provided by M. Quebatte, Biozentrum, Basel), containing the upstream and downstream regions of *rsh* amplified with the following hybridization sequences: 5’-ccggatgatctgaaggaa-3’, 5’-gcgcatcatctgccgaaa-3’ and 5’-gtctgggacctcaagcat-3’, 5’-cccgtggtgacgatatct-3’.

The Δ*rsh* pBBR-*rsh* strain was generated by inserting the pBBR-*rsh* in the Δ*rsh* strain. The pBBR-*rsh* was constructed by cloning the endogenous promoter of *rsh* and the *rsh* coding sequence, amplified with the primers 5’-aaaCTCGAGgcgagattgccgatgaga-3’ and 5’-aaaCTGCAGctatccgttcacacgctttg-3’.

The pBBRi-*mesh1b* strain was constructed by inserting the coding sequence *mesh1b* in the pBBRi plasmid. The sequence of *mesh1b* was adapted to the codon usage of *Brucella*, and is available in Appendix A.

The *mCherry-parB* strains containing pSRK-*relA’* and pSRK-*relA’** were created using the Tn7 system [33] which consists in transposition of mini-Tn7 expressing *mCherry-parB* under the control of the *PgidA* promoter as previously reported [2] and the resistance cassette to ampicillin/carbenicillin under the control of P*bla* promoter at the *glmS* locus of *B. abortus*. The primer sequences used for the amplification of P*gidA*-mCherry-*parB* and P*bla-amp* were 5’-cgcggatcctctgtggaatcctgtttgttg-3’, 5’-AGCGGATACATATTTGAActagctttgaagacggcg-3’ and 5’-TTCAAATATGTATCCGCTCATGA-3’, 5’-cgggatccTTACCAATGCTTAATCAGTGAGG-3’.

### 4.3. Growth Assays

The bacterial growth curves were performed using a bioscreen (Epoch2 Microplate *Photospectrometer* from BioTek). Bacterial cultures in the exponential phase of growth were washed two times with PBS and were normalized at an OD of 0.1 in a given medium. A 200 μL aliquot of the normalized culture was transferred to a plate and each condition was performed in technical triplicate (3× 200 μL). The plate was incubated at 37 °C with shaking and the OD_600_ of each well was measured every 30 min. One biological replicate constitutes the mean of three technical replicates and experiments were repeated at least three times to obtain biological triplicates.

### 4.4. Survival Assays

Bacterial cultures in the exponential phase of growth were normalized at an OD of 0.1 in 2YT liquid medium. Serial dilutions were plated on 2YT solid medium at different time points and plates were incubated at 37 °C.

### 4.5. Infections of RAW 264.7 Macrophages

RAW macrophages were put in wells in DMEM medium (with decomplemented bovine serum, glucose, glutamine, and no pyruvate, Gibco^®^) to have 1 × 10^5^ cells/mL. *B. abortus* 544 was grown in 2YT at 37 °C until exponential phase. The OD of the bacterial culture was measured, and dilutions were performed to have a MOI equal to 50 (50 times more bacteria than macrophages). An input control was performed for each condition by plating bacteria on a 2YT agar plate before infecting cells. Cell medium was removed to add the appropriate bacterial dilution. The mix was centrifuged for 10 min at 1200 rpm (4 °C) and then incubated at 37 °C with 5% CO_2_ (this time point is set as time zero). After one hour of incubation, medium was removed and replaced by medium containing gentamycin (50 μg/mL) for 1 h in order to kill extracellular bacteria, and then by medium containing gentamycin (10 μg/mL). Note that for the experiments using IPTG, the IPTG (10 mM) was kept during all the steps of the infection. At either 2 h, 4 h or 24 h post infection, cells were first washed with sterile PBS and were then incubated in PBS + Triton 0.1% at 37 °C for 10 min in order to lyse the cells while keeping bacteria alive. After that, cells were flushed and lysates were harvested. Serial dilutions were performed and each dilution was spotted on 2YT agar plates and incubated at 37 °C.

### 4.6. Infections of HeLa Cells

HeLa cells were plated in wells in DMEM medium (with sodium pyruvate, non-essential amino acid, glucose, glutamine, and no pyruvate, Gibco^®^) at 4 × 10^4^ cells/mL. *B. abortus* 544 was grown in 2YT at 37 °C until exponential phase, the OD of the bacterial culture was measured, and dilutions were performed to have a MOI equal to 300. An input control was performed for each condition by plating bacteria on a 2YT agar plate before infecting cells. Prior to infections in the presence of IPTG (see below), *relA’* expression was induced 3 h before infection with IPTG (1 mM in YT medium). Cell medium was removed to add the appropriate bacterial dilution. The mix was centrifuged for 10 min at 1200 rpm (4 °C) and incubated at 37 °C with 5% CO_2_ (this time point is set as time zero). After one hour of incubation, medium was removed and replaced by medium containing gentamycin (50 μg/mL) in order to kill extracellular bacteria, and then gentamycin (10 μg/mL). Note that for the experiments using IPTG, the IPTG (10 mM) was kept during all the steps of the infection. At either 2 h, 4 h or 24 h post infection, cells were first washed with sterile PBS and were then incubated in PBS + Triton 0.1% at 37 °C for 10 min in order to lyse the cells while keeping bacteria alive. After that, cells were flushed and lysates were harvested. Serial dilutions were performed and each dilution was spotted on 2YT agar plates and incubated at 37 °C.

### 4.7. G1 Counting

Bacteria in exponential phase of growth were diluted to an OD of 0.1 in 2YT liquid medium with or without IPTG. At each time point, 200 μL of the culture was washed two times in PBS and bacteria were loaded onto a PBS agarose pad to be observed and counted by fluorescence microscopy.

## Figures and Tables

**Figure 1 pathogens-09-00571-f001:**
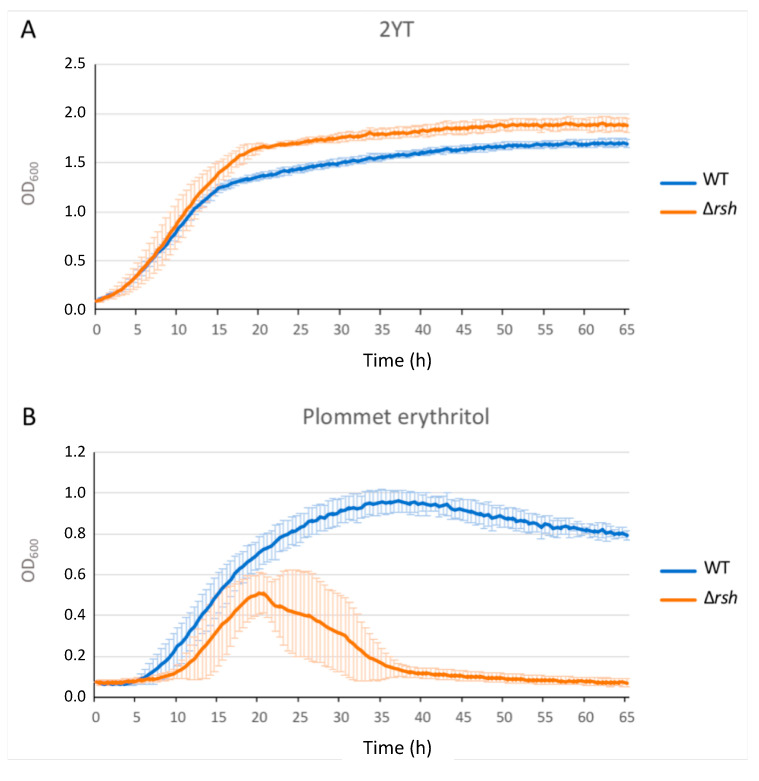
Growth of the Δ*rsh* mutant in 2YT rich medium (**A**) and in Plommet erythritol minimal medium (**B**). Strains were grown in liquid culture overnight in order to reach exponential phase. Cultures were then diluted at an optical density (OD) of 0.1 in 2YT medium. The OD of each strain was measured every 30 min. The graph represents the means of a biological quadruplicate. The error bars represent the standard deviation for each time point. WT: wild type.

**Figure 2 pathogens-09-00571-f002:**
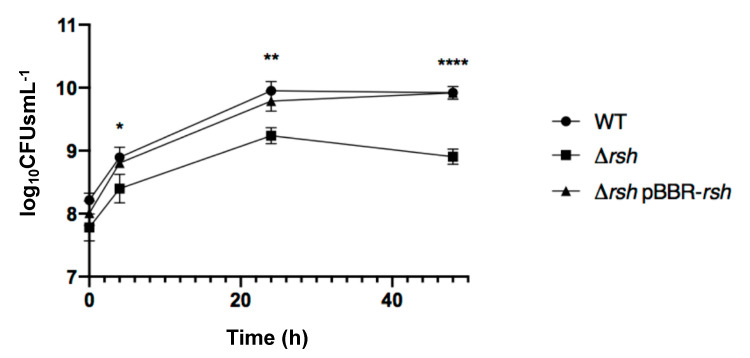
Survival and growth of *B. abortus* WT, Δ*rsh* and Δ*rsh* pBBR-*rsh* in 2YT rich medium. Strains were grown in liquid culture overnight in order to reach exponential phase. Cultures were then diluted at an OD of 0.1 (3 × 10^8^ bacteria/mL for the WT strain) in 2YT medium. The numbers of live bacteria (log_10_CFUs mL^−1^) were determined at 0 h, 4 h, 24 h and 48 h by plating serial dilutions. Values represent the means of three independent experiments and the error bars represent the standard deviation. The asterisks mean significant for *p* < 0.05 (*) *p* < 0.01 (**); *p* < 0.0001 (****), and the *p* values were calculated by one-way ANOVA.

**Figure 3 pathogens-09-00571-f003:**
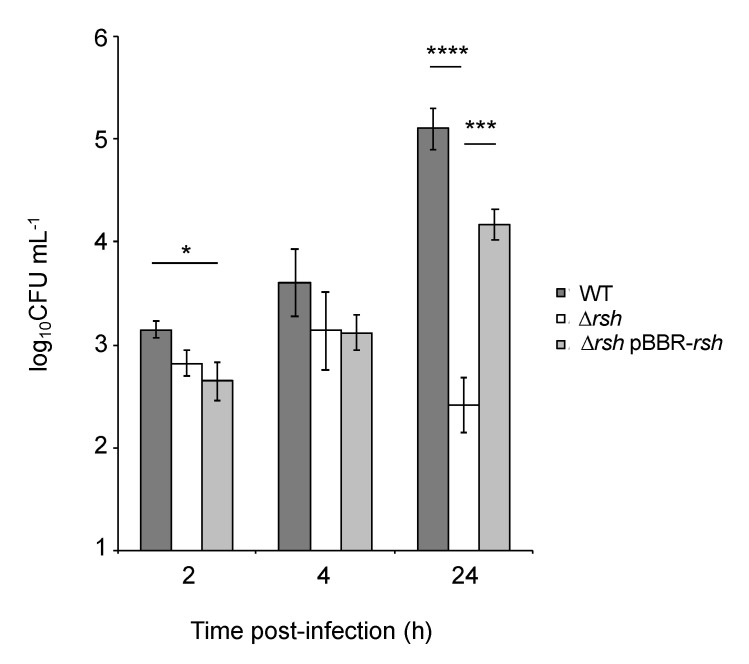
Survival and growth of *B. abortus* WT, Δ*rsh* and Δ*rsh* pBBR-*rsh* during infection of RAW 264.7 macrophages. Strains were grown in liquid culture overnight in order to reach exponential phase. Cultures were then diluted in Dulbecco’s Modified Eagle’s Medium (DMEM) to obtain a multiplicity of infection (MOI) of 50. The numbers of live bacteria (log_10_CFUs mL^−1^ of cellular lysate, 0.5 mL per well) were determined at 2 h, 4 h, and 24 h by plating serial dilutions. Values represent the means of three independent experiments and the error bars represent the standard deviations. A one-way ANOVA test was performed as statistical analysis. The asterisks mean significant for *p* < 0.05 (*); *p* < 0.001 (***); *p* < 0.0001 (****).

**Figure 4 pathogens-09-00571-f004:**
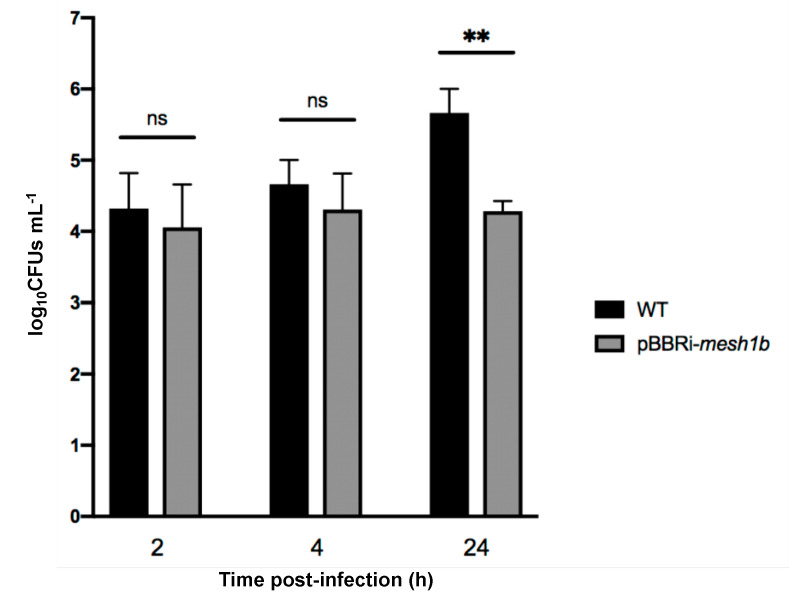
Survival and growth of *B. abortus* WT and pBBRi-*mesh1b* during infection of RAW 264.7 macrophages. Strains were grown in liquid culture overnight in order to reach exponential phase. Cultures were then diluted in DMEM to obtain a MOI of 50. The numbers of live bacteria (log_10_CFUs mL^−1^) were determined at 2 h, 4 h, and 24 h post-infection by plating serial dilutions. Values represent the means of three independent experiments and the error bars represent the standard deviation. A Student *t* test was performed for the comparison of the two strains. The asterisks mean significant for *p* < 0.01 (**) and “ns” means “not significant”.

**Figure 5 pathogens-09-00571-f005:**
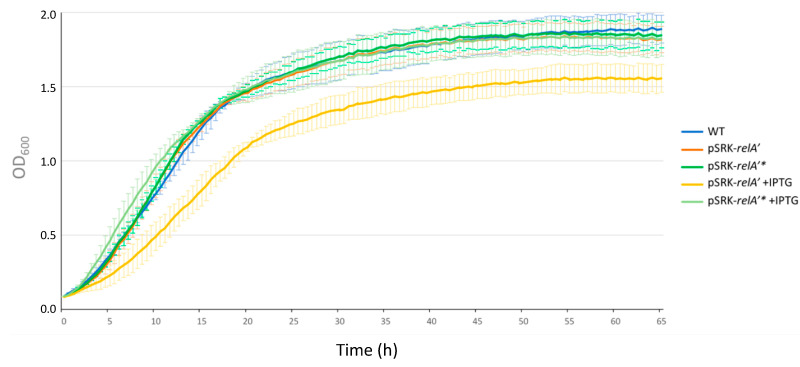
Growth curve in rich culture medium for the WT, *pSRK-relA’* and *pSRK-relA’** with or without IPTG. Strains were grown in liquid culture (2YT medium) overnight in order to reach exponential phase. Cultures were then diluted at an OD of 0.1 in 2YT medium supplemented or not with IPTG. The OD of each strain was measured every 30 min. The graph represents the means of a biological triplicate. The error bars represent the standard deviation for 3 biological replicates for each time point.

**Figure 6 pathogens-09-00571-f006:**
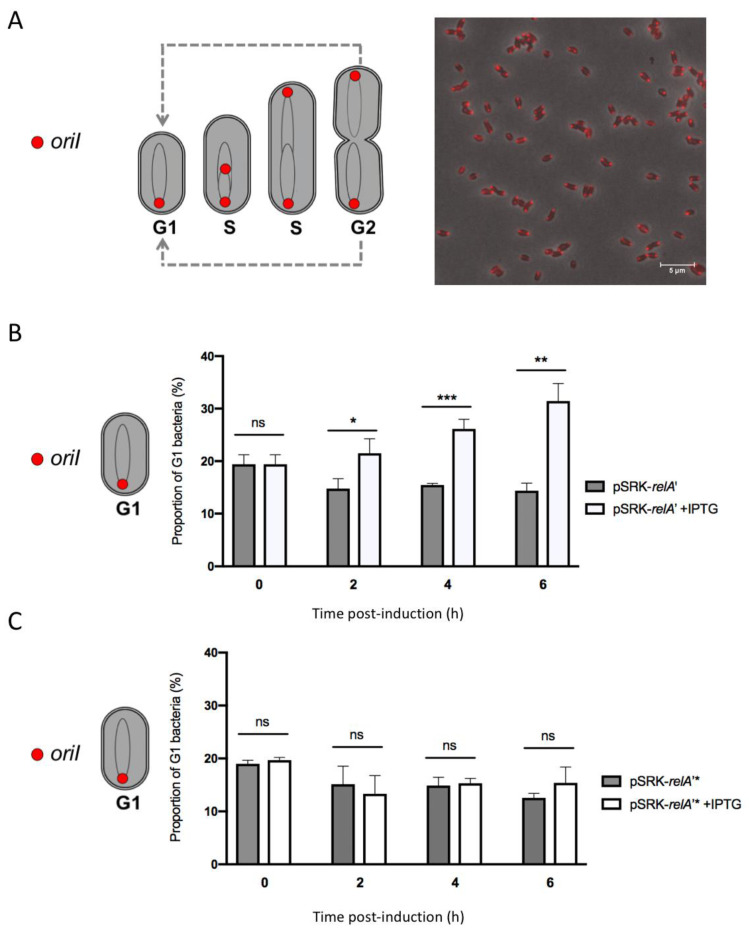
Proportion of G1 bacteria in rich culture medium with or without IPTG for the *pSRK-relA’* and *pSRK-relA’** strains. (**A**) Schematic drawing of the mCherry-ParB localization throughout the cell cycle [2] and fluorescence microscopy of the *pSRK*-*relA’ mCherry-parB* strain. Scale bar represents 5 μm. (**B**) Strains were grown in liquid culture (2YT medium) overnight in order to reach exponential phase. Cultures were then diluted to an OD of 0.1 in 2YT medium supplemented or not with IPTG. Samples were taken every 2 h, placed on a phosphate-buffered saline (PBS) agarose pad and observed with a fluorescence microscope. Bacteria in G1 phase (presenting only one focus of mCherry-ParB) were counted for each time post-induction. Error bars represent the standard deviation from the means of three independent experiments (biological triplicates). The significant differences are indicated by *p* < 0.05 (*), *p* < 0.01 (**) and *p* < 0.001 (***); “ns” means not significant. The number of bacteria considered in these triplicate experiments are detailed in Appendix A.

**Figure 7 pathogens-09-00571-f007:**
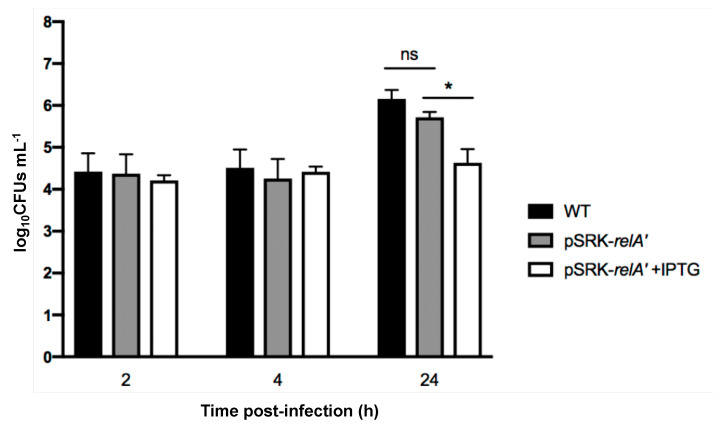
Survival of *B. abortus* WT and *pSRK-relA’* strains with and without IPTG during infection of RAW 264.7 macrophages. Strains were grown in liquid culture overnight in order to reach exponential phase. Cultures were then diluted at an OD of 0.1 with or without IPTG (1 mM) and were incubated for 3 h at 37 °C. Cultures were then diluted in DMEM with or without IPTG (10 mM) to obtain a MOI of 50. Concentrations of live bacteria (log_10_CFUs mL^−1^) were determined at 0 h, 4 h, and 24 h post-infection by plating serial dilutions. Values represent the means of three independent experiments and the error bars represent the standard deviation. A Student’s t test was performed as statistical analysis. The asterisks mean significant for *p* < 0.05 (*) and “ns” means “not significant”.

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
