# Peer review of "Intracellular Growth and Cell Cycle Progression are Dependent on (p)ppGpp Synthetase/Hydrolase in Brucella abortus"

_pathogens, 2020, doi:10.3390/pathogens9070571_

Round 1

Reviewer 1 Report

This study by Van der Henst M et al. shows the role of (p)ppGpp synthetase/hydrolase (Rsh) in controlling the intracellular growth and cell cycle progression of Brucella abortus. The authors perform generally well-controlled experiments and concisely present their results showing the growth-defect of Δrsh mutant both in minimal media and in RAW 264.7 macrophages. Additionally, exogenous expression of Drosophila mesh1 appears to reduce B. abortus survival in RAW 264.7 cells. Further, inducible expression of E.coli relA', expected to overburden the B. abortus cells with (p)ppGpp, induced growth defects suggesting homeostasis of (p)ppGpp within B.abortus is critical for successful infection and proliferation.

This study has various strengths - It is well-written and concise, provides a clear introduction and does not over-analyze the data in the discussion section. My major concerns are in the results section, which are listed below along with other minor corrections as well:

Major comments:

1) There is no biochemical assay showing the functionality of Drosophila mesh1 as a (p)ppGpp hydrolase in B.abortus.

2) For the fluorescence imaging done in Fig. 5, the authors have not provided any fluorescence microscopy images to establish how well their imaging assay is performing and how they are identifying one versus two foci in cells. Please provide some representative images. 

Minor:

1) The survival of Δrsh mutant is more severely affected (Fig. 2B) in comparison to the pBBRi-mesh1b strain (Fig. 3) in RAW 264.7 macrophages. Can you provide some comment regarding this difference in the discussion section? Is this due to some other pro-survival roles of rsh apart from (p)ppGpp regulation?

2) Line 58 - in parenthesis mention that the molecule is guanosine penta- or tetra-phosphate for clarity, since this is the first time (p)ppGpp is introduced. 

3) Line 75 - 'two C-termail' - spelling error?

4) Figure 2B should be Figure 3 instead, since 2A and 2B are different experiments (2YT media versus RAW 264.7 intracellular survival)

5) Line 144-145 - check grammar. 

6) Line 146 - Sun et al 2010 not correctly cited.

7) Line 166- IPTG misspelled. 

Author Response

This study by Van der Henst M et al. shows the role of (p)ppGpp synthetase/hydrolase (Rsh) in controlling the intracellular growth and cell cycle progression of Brucella abortus. The authors perform generally well-controlled experiments and concisely present their results showing the growth-defect of Δrsh mutant both in minimal media and in RAW 264.7 macrophages. Additionally, exogenous expression of Drosophila mesh1 appears to reduce B. abortus survival in RAW 264.7 cells. Further, inducible expression of E.coli relA', expected to overburden the B. abortus cells with (p)ppGpp, induced growth defects suggesting homeostasis of (p)ppGpp within B.abortus is critical for successful infection and proliferation.

This study has various strengths - It is well-written and concise, provides a clear introduction and does not over-analyze the data in the discussion section. My major concerns are in the results section, which are listed below along with other minor corrections as well:

Major comments:

1) There is no biochemical assay showing the functionality of Drosophila mesh1 as a (p)ppGpp hydrolase in B.abortus.

Biochemical assays showing the functionality of Mesh1 have been performed by Sun et al. (2010). They have also shown that Mesh1 was active in Escherichia coli and in HEK293T cells. We use exactly the same coding sequence in our expression plasmid. A sentence was added in the revised manuscript (line 148) to make a clear reference to this work.

2) For the fluorescence imaging done in Fig. 5, the authors have not provided any fluorescence microscopy images to establish how well their imaging assay is performing and how they are identifying one versus two foci in cells. Please provide some representative images.

We agree with reviewer 1 that this should have been added to the original manuscript. A typical picture is now available in Figure 6A, and a supplementary figure (Figure S3) to allow the reader to evaluate the localization pattern of mCherry-ParB.

Minor:

1) The survival of Δrsh mutant is more severely affected (Fig. 2B) in comparison to the pBBRi-mesh1b strain (Fig. 3) in RAW 264.7 macrophages. Can you provide some comment regarding this difference in the discussion section? Is this due to some other pro-survival roles of rsh apart from (p)ppGpp regulation?

A comment is now indicated in the Discussion of the revised manuscript (lines 253-257). It is likely that residual (p)ppGpp is still present in the strain carrying pBBR-mesh1b, while the alarmone would be absent in the ∆rsh strain.

2) Line 58 - in parenthesis mention that the molecule is guanosine penta- or tetra-phosphate for clarity, since this is the first time (p)ppGpp is introduced.

Indeed, it is added in the revised manuscript.

3) Line 75 - 'two C-termail' - spelling error?

Indeed, it is corrected in the revised manuscript.

4) Figure 2B should be Figure 3 instead, since 2A and 2B are different experiments (2YT media versus RAW 264.7 intracellular survival)

We agree with the reviewer. The revised manuscript was modified accordingly.

5) Line 144-145 - check grammar.

In this sentence (now line 146) “generate” was replaced by “generated”, in the revised manuscript.

6) Line 146 - Sun et al 2010 not correctly cited.

The citation is now formatted properly (line 148).

7) Line 166- IPTG misspelled.

IPTG spelling is corrected in the revised manuscript (line 168).

We thank reviewer 1 for his/her comments and corrections.

Reviewer 2 Report

The introduction of this manuscript provides a review of trafficking, cell cycle and regulatory events of Brucella and as compared to some other bacterial pathogens, thus forming the basis for their hypothesis.  The authors performed a logically progressive series of appropriately controlled experiments to test their hypothesis that starvation mediated by (p)ppGpp synthetase/hydrolase regulation of alarmone (p)ppGpp production modulating cell cycle arrest results in the accumulation of more infective Brucella in G1 phase.  The first series of experiments largely confirm previously published results of ∆rsh Brucella and complemented strain responses to stringent liquid culture and intracellular conditions.  Then the authors took their studies further by using codon adapted Drosophila melanogaster mesh1 encoded (p)ppGpp hydrolase resulting in a decrease in Brucella cfu only at 24h post-infection of RAW 264.7 macrophages.  Next they constructed an IPTG-inducible pSRK-relA’ strain of Brucella artificially producing (p)ppGpp and a negative control Brucella catalytically inert strain with a point mutation to indirectly estimate of (p)ppGpp by the amount of limited growth in rich liquid medium, indirectly implying that the quantity of (p)ppGpp production by the induced pSRK-relA’ strain was adequate to reduce growth.  To further characterize the segregation of the replication origins on Chromosome I in the IPTG-inducible pSRK-relA’ strain of Brucella, a mccherry-parB allele was used to measure cells in G1 vs S and G2 phases with or without induction grown in rich medium.  The induced pSRK-relA’ strain of Brucella had a higher proportion G1 bacteria suggesting that the induced production (p)ppGpp reduced the transition from G1 to S and G2 phases within 2 hr.  Infection of RAW 264.7 macrophages with induced vs. non-induced pSRK-relA’ strain and also compared to WT Brucella resulted in reduced intracellular proliferation at 24 hr only in the induced pSRK-relA’ strain.  Finally, to rule out that DksA influences the transcription of the (p)ppGpp, a ∆dksA mutant was found to have no effect on infection or proliferation of Brucella in RAW 264.7 macrophages.  The authors conclude that an appropriate range of the alarmone (p)ppGpp concentration is required for intracellular growth and increasing the proportion of G1 more infectious cells and also that other mechanisms are likely involved in the early control of the intracellular Brucella trafficking and cell cycle.  

While the results of the series of experiments provide good insight suggesting a critical role for (p)ppGpp in intracellular proliferation of G1 arrested Brucella, the inability to measure levels of (p)ppGpp using P32.to confirm the actual concentration of (p)ppGpp under the conditions of these experiments seriously erodes the overall contribution to molecular pathogenesis of B. abortus infection, thus providing the data on (p)ppGpp levels in each of the experiments is highly encouraged.  On the other hand, the results of the experiment do enlighten and provide a framework of future research for increased understanding the role of (p)ppGpp for the intracellular growth of G1 arrested Brucella. 

Author Response

We completely share the frustration of this reviewer about the absence of quantification of (p)ppGpp levels. This is why we multiply the strains in which (p)ppGpp levels are susceptible to be modified by mutation (in the broad sense). Actually, we tried for years to construct a fluorescent reporter for the (p)ppGpp levels, and we failed probably because we cannot control the proteolysis rate in Brucella abortus. Each time a fluorescent reporter was fused to a full length or shortened C-terminal proteolysis tag (which is likely necessary to be able to monitor a decrease of the signal within a reasonable time), the fluorescence of the reporter became undetectable, even if we fused them with a very strong promoter (rrnBP1). In our view, the development of fluorescent reporters with a controlled life-time is crucially needed to set up reporters at the single cell level, but it is a much more complex work than anticipated at the start of these experiments. Indeed, our previous analyses with GFP fusions (Francis et al., 2017, Mol. Microbiol. 103, 780) showed that the variation between individual bacteria is huge, which is adding an additional layer of complexity to these types of analyses. In summary, working with a specific class III pathogen is not similar to the work that could be performed with model organisms or well investigated pathogens like Salmonella, for which numerous tools are available.

Round 2

Reviewer 1 Report

The authors have provided satisfactory responses and enhanced the manuscript considerably. I have no further comments. 

Reviewer 2 Report

I understand the authors' explanations for the lack of biochemical confirmatory assays and find their reasoning acceptable.  I have no additional comments.